



# Performance of an open-path near infrared measurement system for measurements of $CO_2$ and $CH_4$ during extended field trials.

Nicholas M. Deutscher[1], Travis A. Naylor[1], Christopher G.R. Caldow[1,2], Hamish L. McDougall[1], Alex G. Carter[1,†], and David W.T. Griffith[1]

[1]Centre for Atmospheric Chemistry, School of Earth, Atmospheric and Life Sciences, Faculty of Science, Medicine and Health, University of Wollongong, Wollongong, NSW, 2522, Australia
[2]Now at Laboratoire des Sciences du Climat et de l'Environnement, LSCE/IPSL, CEA-CNRS-UVSQ, Universite Paris-Saclay, Gif-sur-Yvette, F-91191, France
[†]deceased, 8 September 2020

**Correspondence:** Nicholas M. Deutscher (ndeutsch@uow.edu.au)

**Abstract.** Open-path measurements of atmospheric composition provide spatial averages of trace gases that are less sensitive to small-scale variations and the effects of meteorology. In this study we introduce improvements to Open-Path Near Infrared (OP-NIR) Fourier Transform Spectrometer measurements of $CO_2$ and $CH_4$. In an extended field trial, the OP-NIR achieved measurement repeatability six times better for $CO_2$ (0.26 ppm) and ten times better for $CH_4$ (2.0 ppb) over a 1.55 km one-way

path than its predecessor. The measurement repeatability was independent of pathlength up to 1.55 km, the longest distance tested. Comparisons to co-located in situ measurements under well-mixed conditions characterise biases of 2.67% for $CO_2$ and 2.46% for $CH_4$ relative to in situ measurements calibrated to World Meteorological Organisation - Global Atmosphere Watch (WMO-GAW) scales. The OP-NIR measurements can detect signals due to local photosynthesis and respiration, and local point sources of $CH_4$. The OP-NIR is well-suited for deployment in urban or rural settings to quantify atmospheric composition on

kilometre scales.

## 1 Introduction

Anthropogenic greenhouse gases, primarily carbon dioxide ($CO_2$) and methane ($CH_4$), are the main drivers of systematic changes in the Earth's climate, especially a net warming of the atmosphere and oceans. Characterising sources and sinks of

15 greenhouse gases to and from the atmosphere is critical to understanding past, present, and future change.

Open-path measurements of atmospheric composition can provide complementary information to that available from in situ point measurements. By integrating over kilometre-length scales, open-path measurements provide quantifications that are less sensitive to small-scale variability and more representative of model spatial scales. In applications around detecting remote



point sources, an integrated path also improves the likelihood of sampling through an emission plume. These advantages may
come at the cost of precision and accuracy compared to in situ point measurements.

Technologies available for open-path measurements include differential optical absorption spectroscopy (DOAS), tunable
diode laser absorption spectroscopy (TDLAS), frequency comb spectroscopy, and Open-Path FTIR (OP-FTIR).

DOAS is typically applied in the ultraviolet and visible regions of the spectrum, making it unsuitable for measuring green-
house gases. Recent advances have seen the expansion of DOAS-type measurements into the near infrared (NIR), delivering
repeatability in $CO_2$ measurements of 2-4 ppm over km-scale pathlengths, and $CH_4$ repeatability on the order of 200 ppb
(Queisser et al., 2016; Saito et al., 2015).

Tunable diode lasers are a relatively inexpensive technology capable of measurements of many atmospheric constituents,
depending on the internal laser used. These have been applied to the detection of $CO_2$ and $CH_4$ in the past, but measured
amounts vary considerably from instrument to instrument, and they are susceptible to drifts and inaccuracies (Feitz et al.,
2018). In general, laser-based techniques can have low power requirements, but are only tuned around a single wavelength, and
therefore can typically only detect a single gas. In addition this leaves them susceptible to potential interferences. GreenLITE™
(Dobler et al., 2013; Lian et al., 2019) gets around some of the interference aspect by measuring at two wavelengths.

Frequency comb spectroscopy enables measurements across a range a spectral wavelengths. For example, Alden et al. (2019)
employ a dual frequency comb spectrometer to measure methane, operating over a 270 $cm^{-1}$ spectral window centred at 7000
$cm^{-1}$ at high spectral resolution (0.0067 $cm^{-1}$). This combination of a wider window and high spectral resolution allows
for the simultaneous retrieval of interfering species, especially atmospheric water vapour, improving the inherent stability and
accuracy of the measurement. The instruments can operate in different wavelength regions; Rieker et al. (2014) describe an
earlier system operating around 6000 $cm^{-1}$ at 0.0033 $cm^{-1}$ resolution, which measures $CO_2$, $CH_4$, $H_2O$, HDO, and $^{13}CO_2$
over 2km pathlengths, with 5 min precision of better than 1 ppm for $CO_2$ and 3 ppb for $CH_4$.
While not operating at the spectral resolutions achieved by laser-based spectroscopy, open-path FTIR spectrometers can
operate over even wider spectral regions, allowing quantification of many gases that absorb infrared radiation in the atmosphere.
Open-path mid-infrared (OP-MIR) measurements (200 - 4000 $cm^{-1}$) are limited to relatively short pathlengths because of the
low brightness of the internal MIR light source. Open-path MIR measurements have been used to quantify $CO_2$ and $CH_4$ on
paths up to 400m (e.g. (Desservettaz et al., 2019; Phillips et al., 2019)) in an urban environment. More typically they have
been applied on 100m scales, for quantification of area sources coupled with a tracer gas, micrometerological techniques or a
backward lagrangian stochastic model (WINDTRAX) (Bai et al., 2018; Flesch et al., 2016; Jones et al., 2011; Laubach et al.,
2013; Loh et al., 2009; Naylor et al., 2016; Phillips et al., 2016). OP-MIR has also been used to detect and quantify a point
$CO_2$ and $CH_4$ source during a controlled release experiment (Cartwright et al., 2019; Feitz et al., 2018). At these pathlengths,
repeatability of 1 ppm $CO_2$ and 5 ppb for $CH_4$ can be achieved. Tomographic techniques for quantifying fluxes, such as
described by Humphries et al. (2012), could be applied to open-path measurements.

Griffith et al. (2018) detailed a pilot study extending OP-FTIR to measurements in the near infrared (NIR) spectral region
(4000 - 13,500 $cm^{-1}$), where higher source brightness results in better collimation of the beam, and theoretically longer paths.
In addition, there are fewer interfering species, though the absorptions of $CO_2$ and $CH_4$ are weaker relative to MIR as they are



overtone and combination bands. The instrument was deployed over a 1.55km one-way path, and demonstrated repeatability
of 1.7 ppm for $CO_2$ and 21 ppb for $CH_4$ for 5 minute averages. This manuscript details further developments and field trials
based on the instrument introduced in that study, and demonstrates significant improvements in precision and accuracy.

Section 2 describes the instrument, data processing and corrections applied to the data. The field deployment and testing of
the instrument is described in Section 3, while the results are presented and discussed in Section 4. The conclusions are then
presented in Section 5.

## 2  Open-path Near InfraRed Spectrometer

The instrument described here is based on the instrument described by Griffith et al. (2018), which was deployed in Heidelberg,
Germany. We describe the instrument, retroreflector array and data processing, and highlight the modifications relative to
Griffith et al. (2018).

Figure 1 shows a schematic of the instrument setup. The central component of the open-path NIR (OP-NIR) system is a
low spectral resolution FTIR spectrometer, the Bruker IRcube (Bruker Optik, Ettlingen, Germany). The spectrometer has an
internal 25W tungsten globe restricted to 11W as NIR radiation source, quartz beamsplitter and TE-cooled Indium Gallium
Arsenide (InGaAs) detector optimised for the NIR spectral region, covering the spectral range from 3800 to 10,000 $cm^{-1}$.
This covers the same spectral range as that used in the Total Carbon Column Observing Network (TCCON) (Wunch et al.,
2011), which measures total column $CO_2$, $CH_4$ and other gases using the sun as a light source. Unlike TCCON, the IRcube
used in this study has a low spectral resolution of 0.5 $cm^{-1}$ (maximum optical path difference of 1.8 cm). The spectrometer is
relatively compact (310 x 290 x 220 mm, 14 kg).

For this work, the relatively small spectrometer allows it to be mounted on an Automated Instrument Mount (AIM, Omegalec,
Unanderra, NSW, Australia), previously also used for open-path MIR measurements. AIM allows programmable, adjustable
pointing of the instrument, driven by stepper motors on each of a elevation and azimuth axis. AIM and the instrument are
supported on a heavy-duty tripod (MOOG-QuickSet, Northbrook, IL, USA). The spectrometer is rigidly coupled to a beam-
expanding telescope. In contrast to the setup used by Griffith et al. (2018), here we use a 50:50 zinc selenide (ZnSe) beamsplitter
instead of an optic fibre to couple the interferometer and detector to the telescope, and a Schmidt-Cassegrain (Meade Instru-
ments Corporation, Irvine, CA, USA), instead of a Newtonian, telescope. In addition, the light is modulated prior to exiting the
spectrometer, minimising the effects of stray light from the environment on the measurements. The telescope has the Schmidt
corrector plate removed and uses a custom convex 30 mm diameter secondary mirror with 50 mm focal length mounted on a
micrometer stage for fine focus adjustment to convert the telescope to a beam expander. Direct back reflection off centre of the
secondary mirror to the beamsplitter, which would result in a shorter measurement path, is minimised by masking the centre of
the secondary mirror using non-reflective material. This setup more closely resembles that used with existing OP-MIR systems
(Flesch et al., 2016; Naylor et al., 2016; Phillips et al., 2016), and in initial lab and field trials provided an order of magnitude
more throughput than the optic fibre coupling described by Griffith et al. (2018).



The retroreflector initially used was a hollow gold-coated corner cube array (30 x 63mm, PLX Inc, Deer Park, NY, USA) previously deployed with the OP-MIR systems. This was later replaced by an array of 49 x 50mm solid BK-7 glass retroreflectors (JEOC, Zhejiang, China). The solid glass corner cubes are less prone to condensation and dust accumulation, easily cleaned, and require less maintenance in the field, as well as being considerably less expensive to purchase. The beam transmitted over

90 the open path is returned to the telescope via the retroreflector array. The returning beam is reflected off the beamsplitter, and focussed onto the InGaAs detector in the spectrometer.

Auxiliary information is logged at the spectrometer end of the path. A temperature sensor (LM335 diode) is housed in a solar shield to protect from direct solar radiation, while a pressure sensor (Vaisala PTB110, Helsinki, Finland) is housed in a vented box. These are logged on the same time interval as spectral collection.

The instrument is powered by 240V mains power. For field deployment, it was housed in a portable aluminium garden shed.

## 2.1 Data processing

From the collected NIR spectra, path-averaged trace gas mole fractions are retrieved by fitting a calculated spectrum to the measured spectrum. The calculated spectrum is generated using a forward model, Multiple Atmospheric Layer Transmission (MALT, Griffith (1996)). The forward model uses absorption line parameters based on HITRAN 2008 (Rothman et al., 2009),

measured temperature, pressure and pathlength, and an initial estimate of trace gas amounts, together with an instrument model accounting for resolution, apodisation, lineshape, spectral shift and continuum shape. The calculated spectrum is iteratively adjusted using the Levenberg-Marquardt algorithm to achieve the best fit to the measured spectrum. This yields a path-integrated number of molecules (column density, molecules $cm^{-2}$), which is converted to a path-averaged mole fraction using the air density calculated from pressure and temperature. Spectra are fitted in a number of different windows. Those used for this

work are summarised in Table 1 and are illustrated in Griffith et al. (2018).

Spectra are fitted in real time immediately after collection to provide an initial estimate of trace gas mole fractions; however, we subsequently re-fit the spectra to account for two corrections: 1) the residual back-reflection from the secondary mirror; and 2) differences between the temperature measured at one end of the path and the effective or true path-averaged temperature.

### 2.1.1 Short-path correction

A small fraction of the outgoing beam is reflected directly back from the centre of the convex secondary mirror to the beamsplitter and detector. Outgoing radiation can also be reflected by the external beamsplitter and returned to the beamsplitter and detector after diffuse reflection from the inside of the instrument cover. Both of these effects result in radiation that does not traverse the full atmospheric path. These effects are minimised by blocking the centre of the secondary mirror and shielding the instrument cover with a non-reflective, matt black surface. Despite these measures, the short-path signal is not completely

eliminated, but remains relatively constant and is independent of the long-path intensity. It therefore represents a variable relative contribution to the total intensity.

For a more accurate analysis of the long-path spectrum, a correction for this effect is derived and applied. An off-target spectrum is recorded and stored by pointing the telescope away from the retroreflector to ensure there is no long-path contribution.



A least-squares fit of the stored off-target spectrum is performed for each long-path measurement in the 5000 - 5500 cm$^{-1}$
region, which is otherwise blacked out by H$_2$O absorption in the full path. The least-squares fit provides a scaling factor to
apply to the stored off-target spectrum, which is then subtracted from the measured spectrum across the full wavelength range.
The spectral analysis is then repeated on the resulting long-path only spectrum.

### 2.1.2 Path-averaged temperature

Temperature is required in the spectral analysis to generate the forward model spectrum from HITRAN line parameters, and
to calculate air density when deriving mole fractions. Air density is directly inversely proportional to temperature, therefore a
1% error in absolute temperature will result in an inversely proportional error in mole fraction. The measured temperature at
one end of the path is not necessarily representative of the effective mean temperature along the long-path due to variation of
the temperature within the path and the possibility that the measured temperature could be influenced by solar radiation and
thermal mass of surrounds. In general, the measured temperature is also likely to be made closer to the ground than the average
measurement height (beam path) along the open path.

To determine the effective path-averaged temperature, an initial fit to the CO$_2$ window (Table 1) is performed including the
temperature in the least-squares fit. The retrieved temperature is then applied as a fixed value in reanalysis of all windows. The
retrieved temperatures show the greatest difference from the measurements during daytime, when solar heating of the sensor
results in a positive bias in the apparent measured values and can be as much as 5-10 °C. Waxman et al. (2017) found that a
similar process was necessary with frequency comb spectroscopy open-path measurements.

The net effect of the refitting process to account for both the short-path correction and path-averaged temperature yields the
differences illustrated in Figure 2. The refitted temperatures are generally lower. This, together with removing the shortpath
absorption, results in an increase in retrieved mole fractions after refitting.

## 3 Field deployment

After initial testing over short pathlengths within the lab and on the University of Wollongong campus, the instrument was
deployed on an extended field campaign to a New South Wales Department of Primary Industries (NSW-DPI) agricultural
research site, Elizabeth Macarthur Agricultural Institute (EMAI) at Menangle, NSW. The site is located just outside the current
Sydney urban area to the south-west of the city. The location is depicted in the map in Figure 3.

The instrument was located on site for about four months (123 days), from August 4, 2018 to December 12, 2018. Through-
out the deployment period, several changes to the setup took place, as summarised in Table 2. Firstly, the pathlength was varied
by moving the retroreflector array from 0.6 km to 1.11 km to 1.55 km distant from the FTIR, resulting in total pathlengths of
1.20, 2.22 and 3.10 km. The terrain limited the one-way path to a maximum of 1500 m. At 600 m, the retroreflector array was
changed from the hollow gold-coated array to the array of BK7 glass corner cubes. Finally, the telescope was changed from a
10-inch to 12-inch version.



Accounting for two periods where the instrument was removed for 1) a re-alignment; and 2) the change in telescope size, the OP-NIR measured successfully during the remote deployment for 92% of the possible time. Data losses occurred due to power failures and on site instrument maintenance. After filtering to remove spectra with small signal due mostly to rain, fog and condensation on the retroreflectors (less than 5% of the maximum) and poor fits (RMS residual greater than 0.007), a further 9% of measurements were removed, mostly due to moisture, meaning the system was able to provide measurements of

path-averaged mole fractions 83% of the time.

### 3.1    Cavity RingDown Spectrometer (CRDS) in situ measurements

During the campaign, a Picarro G2301 Cavity RingDown Spectrometer (CRDS) (Crosson, 2008), measuring $CO_2$, $CH_4$ and $H_2O$ was deployed. The CRDS measured continuously and was co-located with the OP-NIR spectrometer from November 30 to December 12, 2018. The CRDS sampled from an inlet mounted above the roof of the instrument shed at approximately

2.5m above ground level. It was calibrated against a suite of WMO-traceable gas standards before and after the deployment and showed no drift in calibration during the period.

### 3.2    Meteorological data

Weather data are collected on site by NSW-DPI 1.2 km to the south-south-east of the OP-NIR instrument shelter. These are collected with 10 min resolution using a Measurement Engineering Australia all-in-one weather station, supplying wind speed

and direction, temperature, humidity, solar radiation, and precipitation.

### 4    Results

The time series of $CO_2$, $CH_4$, $O_2$. $H_2O$, CO and $N_2O$ from the EMAI deployment are shown in Figure 4 for the 1110 and 1500 m pathlengths. While the CO ($120 \pm 80$ ppb (mean $\pm$ standard deviation)) and $N_2O$ ($330 \pm 60$ ppb) retrievals are centred on realistic values, the scatter about these is large due to the transmission cutoff of the BK7 glass corner cubes at lower frequencies

where these species absorb. This could be improved by using corner cubes of different glass or quartz, but the focus of the remainder of this work will be on $CO_2$ and $CH_4$.

### 4.1    Measurement performance

We assess the performance of the system using a variety of metrics. Firstly, we calculate the signal-to-noise ratio (SNR) within the spectra by ratioing consecutive spectra and calculating the root-mean-square (RMS) noise of the result in the 6300 - 6500

$cm^{-1}$ spectral region, free from atmospheric absorption.

     Secondly, we assess the repeatability of the path-averaged mole fraction measurements by choosing periods when atmospheric variability is small, specifically at windspeeds greater than 2 $ms^{-1}$. This does not eliminate all sources of real atmospheric variability, therefore a low frequency running mean (40 points/2 hours) was also subtracted from the time series in order



to assess the variability in the trace gas measurements. Allan deviations (Allan, 1966; Werle et al., 1993) are also calculated,
and for single measurements agree with the standard deviations of the noise time series.

The results are summarised in Table 3 for varying pathlengths and instrument setups with the results from the original
Heidelberg deployment (Griffith et al., 2018) for comparison. A change of retroreflector array at 600m pathlength yields SNR
improvement of about a factor of three, and a small improvement in trace gas retrieval repeatability for $CO_2$ and $CH_4$, as
shown in the difference in performance between measurement periods 1 and 2. The retroreflector arrays are nearly identical in
size, therefore this is attributed to the newer, cleaner optics. Measurements of CO and $N_2O$ are, however, compromised by the
change in reflector due to lower transmission at these wavelengths through the glass corner cubes.

For a detector-noise-limited spectrum measurement, increasing pathlength from 600m to 1110m to 1500m would decrease
spectrum SNR as the inverse square of the pathlength while the depth of absorption lines increases in proportion to pathlength.
In practice (Table 3) we observe a fall in spectrum SNR with pathlength that is approximately linear and which we interpret as
due to a component of signal-dependent noise due to atmospheric turbulence (and scintillation) near the ground. Under these
conditions the measurement precision is essentially independent of pathlength, as observed.

An increased telescope diameter from 10" to 12" at 1550 m also yielded a 40% improvement in SNR, compared to a
theoretical increase of 44% based on the relative increase in surface area. Larger telescope diameters should therefore yield
further improvements in SNR. This setup yielded the best trace gas repeatabilities, though some variability between setups
occurs from the availability of suitable well-mixed atmospheric periods and underlying atmospheric variability.

At almost identical pathlength, the new system has four times better SNR and repeatability six times (0.26 ppm) and ten
times (2.0 ppb) better for $CO_2$ and $CH_4$, respectively, than the original system described by Griffith et al. (2018).

## 4.2 Comparison to in situ measurements

The time series of OP-NIR and CRDS measurements and their differences are shown in Figure 5. In the majority of the
time series the two instruments capture the same broad events and the OP-NIR compares reasonably well with the Picarro
measurements over the concurrent measurement period. The OP-NIR measurements typically exhibit less variability than the
in situ measurements, especially at night and/or during low windspeed conditions. This is expected, because the OP-NIR
measures the spatial average of gas mole fractions, and thus variability, along the path averaged over the time of each OP-NIR
measurement - 3 minutes in this case. In contrast, in situ techniques are sensitive to spatio-temporal variability at their location
on the time scale of several seconds, such as localised gas fluxes and/or atmospheric mixing or lack thereof.

The comparison between the OP-NIR and Picarro for $CO_2$ is shown in Figure 6. The measurements are divided into three
categories: sunny (solar radiation above 100 W m$^{-2}$), well-mixed (wind speed greater than 2 m s$^{-1}$) and all other data.
Picarro $CO_2$ measurements are consistently higher at night and lower during the day in sunny periods compared to the OP-NIR
measurements. Usually over grassland such as at the field site, there is a net flux of $CO_2$ from the atmosphere to the biosphere
during the day, due to photosynthetic uptake being larger than respiration. The reverse is true at night. These fluxes establish
a vertical $CO_2$ gradient near the Earth's surface, with higher $CO_2$ near the surface during the night, and lower $CO_2$ near the
surface during the day. The CRDS inlet is closer to the surface (2.5 m) than the mean height of the OP-NIR path (23 m,





see Figure 7), which may explain why the Picarro $CO_2$ measurements are higher at night and lower during the day than the OP-NIR.

215 For $CH_4$ (Figure 8) variations in both directions occur with the OP-NIR measurements often higher than the CRDS. This is due to the presence of two local sources in the fields over which the open-path beam passes - a coal-seam gas (CSG) well located almost directly under the path at 500m from the retroreflector, and grazing stock (sheep and cattle) most often located around the eastern end of the path. The CRDS is located to the western side of the field and since it is a point rather than path-integrated measurement, it therefore samples less influence from these sources.

220 A polar bivariate plot for $CH_4$ is shown for the data from the entire campaign in Figure 9. Highest $CH_4$ values are seen under conditions with low wind speeds, indicating a source under or very close to the measurement path and/or a build-up of $CH_4$ under stable atmospheric conditions. Other high amounts are also seen from a sector to the south east, with some enhancements to the north. The nearest CSG well is located approximately 5m to the south of the open-path, 500m from the retroreflector end of the 1500m path. When they were in nearby fields, the majority of sheep were located to the east/south-east of the instrument

225 shed (south of the retroreflector), while cattle were periodically located to the south-east of the instrument, and regularly to the north. As the experiment was conducted remotely, we do not have exact records of stock movements.

 For $CH_4$, an additional criterion is therefore added to the definition of well-mixed conditions to account for the CSG well, sheep and cattle that are present near to the OP-NIR measurement path. We therefore also include only data from the westerly sector in our definition of well-mixed conditions; i.e. wind directions between 180 and 360 degrees.

230 Comparison between the in situ CRDS measurements and the path-averaged FTIR measurements under well-mixed conditions is used to assess the bias of the OP-NIR system. Under these conditions, the relationship between the OP-NIR and CRDS falls tightly on a line. For $CO_2$, this corresponds to a slope of $1.0267 \pm 0.0002$; i.e. the OP-NIR is biased high by 2.67% ($\approx$10 ppm at 400 ppm) relative to the in situ measurement scale. Figure 8 shows the comparison between in situ and open-path $CH_4$. With wind speeds above 2 m s$^{-1}$ from the west, the relationship between the open-path and in situ measurements lies around

235 a slope of $1.0246 \pm 0.0004$. There is much more scatter in the relationship, indicative of the variety and spatial inhomogeneity of local sources and sinks.

 The comparison between the OP-NIR and in situ measurements is summarised in Table 4.

### 4.3 Future Directions

At EMAI, we were limited to testing up to 1.55 km due to the availability of appropriate lines-of-sight, but the pathlength-

240 independent behaviour of the OP-NIR measurement repeatability out to 1500 m suggests that longer pathlengths are possible. Given the cost-effective BK7 glass retroreflectors, the surface area of the retroreflector array could be further increased to facilitate this. A larger telescope could also increase signal, enabling longer pathlengths to be used. The inexpensive nature of the retroreflectors, and the use of the AIM unit with programmable pointing, means that a multiple path configuration would also be possible. This could produce a doubling of effective pathlength or enable setups where upwind/downwind measurement

245 pairs can be taken over wider areas than previously possible with the OP-MIR system (Cartwright et al., 2019; Feitz et al., 2018).

The ability to co-retrieve CO and $N_2O$ would enable the system to be used in a broader range of applications, in both urban and rural settings. At moderate increased expense, alternative glass retroreflectors could be used to improve signal at these wavelengths, which should lead to improved precision. These co-retrieved gases could help to identify and quantify atmospheric signals especially from combustion (CO) or soil/wastewater processes ($N_2O$).

Open-path measurements are more likely to capture point source emissions than single point measurements provided that the repeatability is good enough to resolve this over the integrated path. The OP-NIR spatial scale is more comparable to the scale of many atmospheric models, which facilitates model intercomparison studies. Other potential applications include measurements from industrial scale composting, landfills, and wastewater treatment, agricultural monitoring of soil (fertiliser) or animal emissions, and gas pipeline monitoring and leak detection when combined with appropriate modeling.

## 5 Conclusions

In this paper, we present refinements to an open-path Fourier Transform Infrared system measuring in the near infrared spectral region. Over paths up to 1.55 km one-way, the system can achieve repeatability of better than 0.1% (at 0.26 ppm) for $CO_2$ and close to 0.1% (at 2.0 ppb) for $CH_4$ for 3 minute averaging times under well-mixed atmospheric conditions. The measurement precision is essentially independent of pathlength. Comparison to co-located in situ measurements, also under well-mixed conditions, indicates offsets of approximately 2.5% from measurements traceable to WMO scales. The open-path measurements therefore have a calibration offset of 2.67% for $CO_2$ and 2.46% for $CH_4$. The open-path measurements also prove capable of detecting signals from photosynthesis, respiration, and ruminant stock on local scales.

*Author contributions.* NMD led the research and co-ordinated writing of the paper. NMD, and DWTG were responsible for acquiring funding for this research. NMD, DWTG and TAN organised field trials, which were executed by them, CGRC and HLM. NMD, DWTG, TAN and CGRC were involved in project direction. TAN, DWTG, NMD, CGRC and AGC were all involved in data analysis. All authors contributed to writing and editing the manuscript.

*Competing interests.* The authors declare no competing interests.

*Acknowledgements.* The authors gratefully acknowledge staff at Elizabeth Macarthur Agricultural Institute, especially Greg Scott and Ania Deutscher for organising and providing access to the EMAI field site and Michael Fitzgerald for providing weather station data. The authors wish to acknowledge financial assistance provided through Australian National Low Emissions Coal Research and Development (ANLEC R&D). ANLEC R&D is supported by COAL21 Ltd and the Australian Government through the Clean Energy Initiative. This research project used assets provided by the Australian Government Education Investment Fund through the CO2CRC. We also acknowledge the support of





AuScope for the Picarro CRDS analyser used in this work. NMD is funded via an ARC Future Fellowship, FT180100327. We an indebted
to the input of colleagues throughout the project, especially Peter Rayner and Jeremy Silver at the University of Melbourne.



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





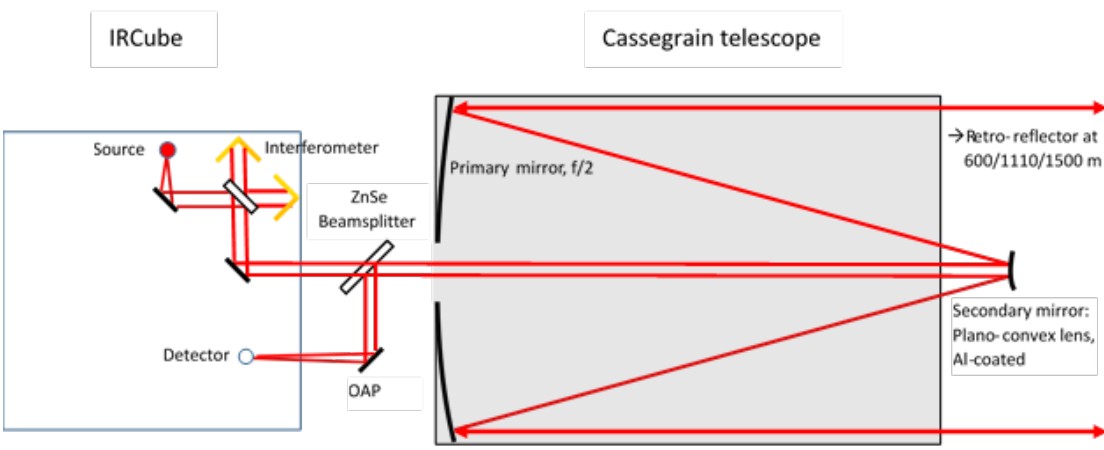

**Figure 1.** Schematic diagram of the setup of the OP-NIR spectrometer, its coupling to the Cassegrain telescope, and the lightpath through these and subsequent to/from the retroreflector.

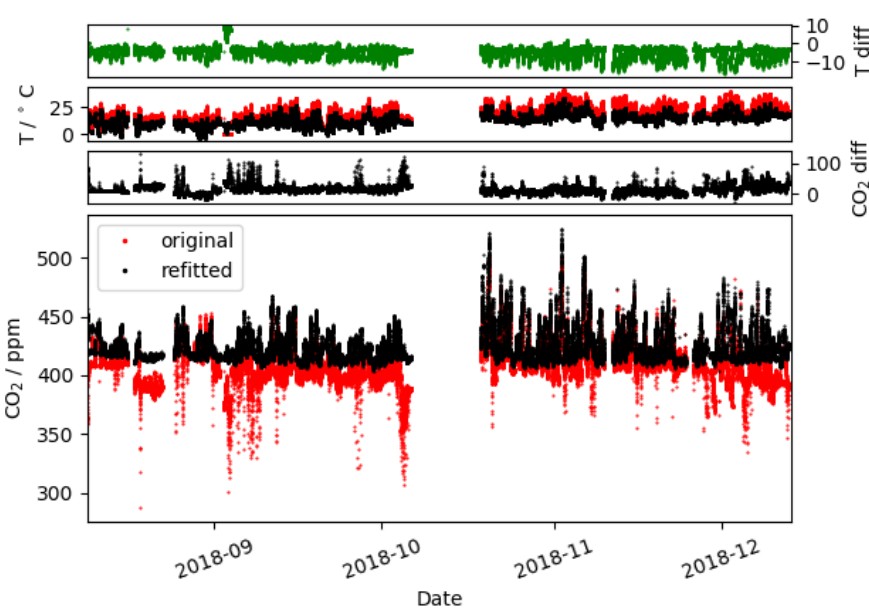

**Figure 2.** Timeseries illustrating the impact of re-processing the original spectra to account for contributions from short-path (spurious) reflections and inaccuracies in measured temperature. The figure shows in order from bottom to top: the original (red) and refitted (black) $CO_2$ (bottom), the difference between the refitted and original $CO_2$ (2nd from bottom), the measured (red) and fitted (black) temperature (2nd from top), and the difference between the fitted and measured temperature (top).



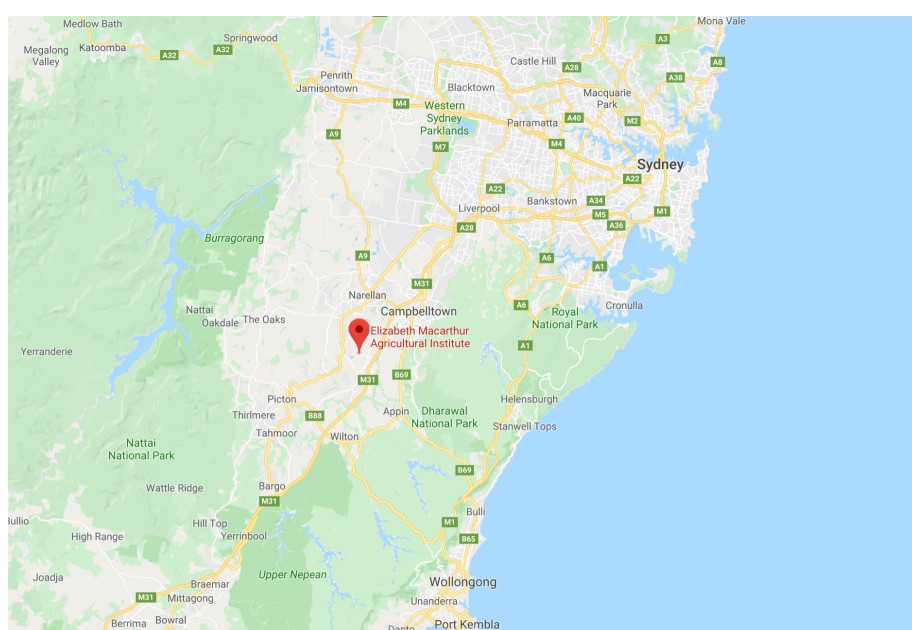

**Figure 3.** Map showing the location of EMAI relative to Sydney and Wollongong. The measurement path and corresponding elevation profile are shown in Figure 7. Map data © Google Maps.



**Figure 4.** Time series of the retrieved path-averaged mole fractions for the 1110m and 1500m pathlengths. Data collected at a one-way pathlength of 1110m are shown in grey, and the 1500m pathlength in black. The $CH_4$ y-axis has been zoomed to 1700 to 2500 nmol mol$^{-1}$.





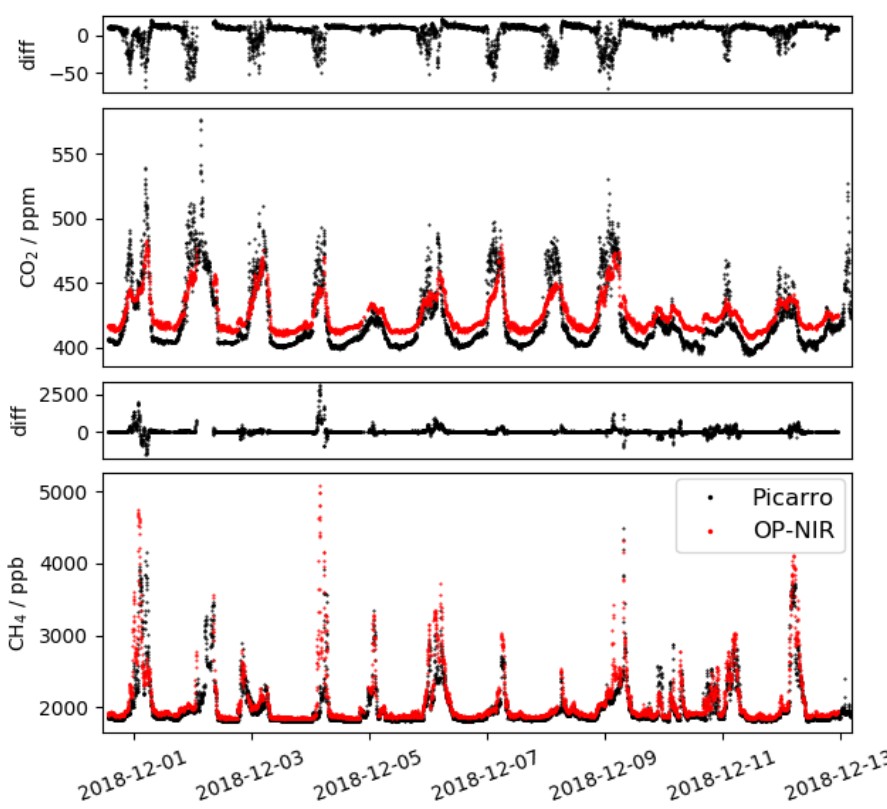

**Figure 5.** Time series of measured $CO_2$ (top) and $CH_4$ (bottom) from the OP-NIR (red) and in situ CRDS (black) measurements while the CRDS was co-located with the OP-NIR, from December 1-13, 2018. For each gas, the upper panel shows the difference between the OP-NIR and CRDS measurements.





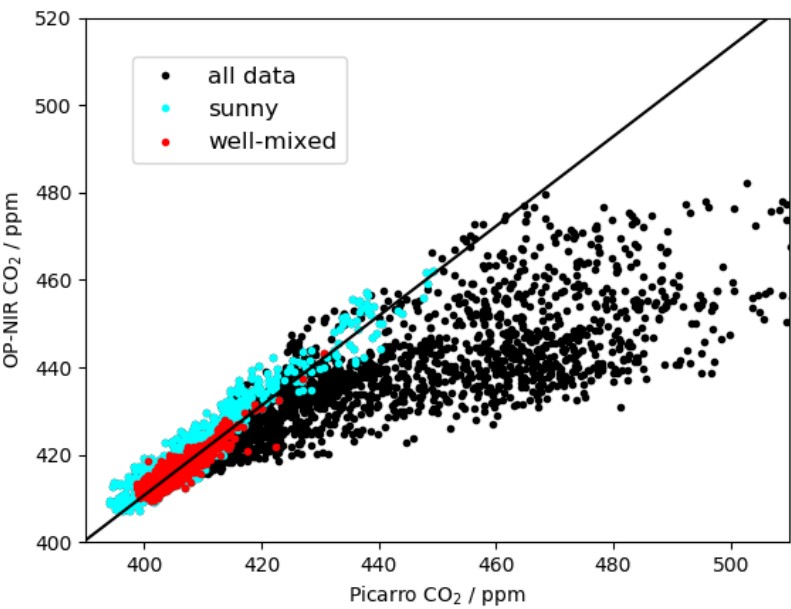

**Figure 6.** Comparison of path-averaged $CO_2$ measured by the open-path NIR and in situ $CO_2$ measured by the CRDS. All data points are shown in black, with those from well-mixed conditions (wind speeds greater than 2 m s$^{-1}$) in red, and those during sunlight hours (solar radiation greater than 100 W m$^{-2}$) in cyan. The best fit line (OP-NIR = $1.0267 \pm 0.0002$ * Picarro) for the fit of OP-NIR vs in situ measurements under well-mixed conditions is shown.

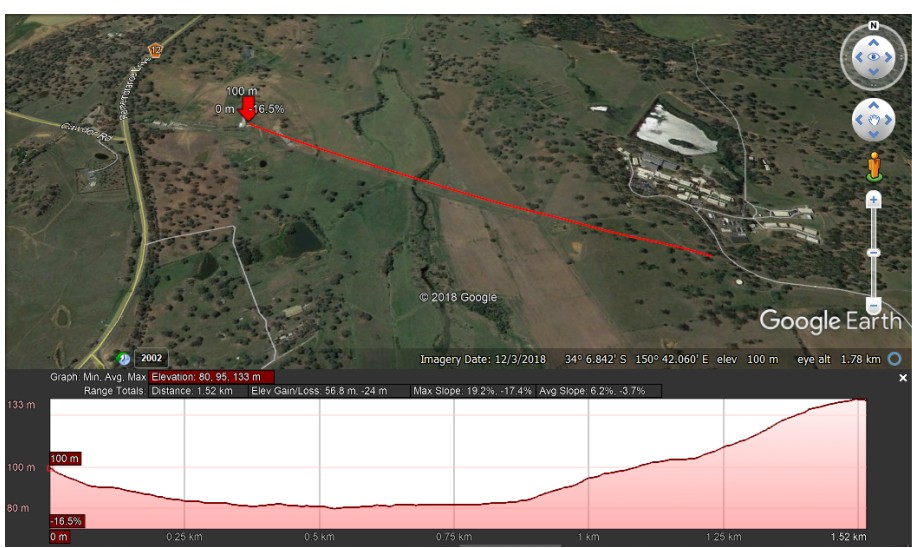

**Figure 7.** Map of the 1.55 km path at EMAI and corresponding elevation profile. The OP-NIR and stationary Picarro were located at the left end of the path and the retroreflector array at the right end. The mean elevation across the 1.55 km observation path is 118 metres above sea level an average of 23 metres above surface elevation. Map © Google Earth.



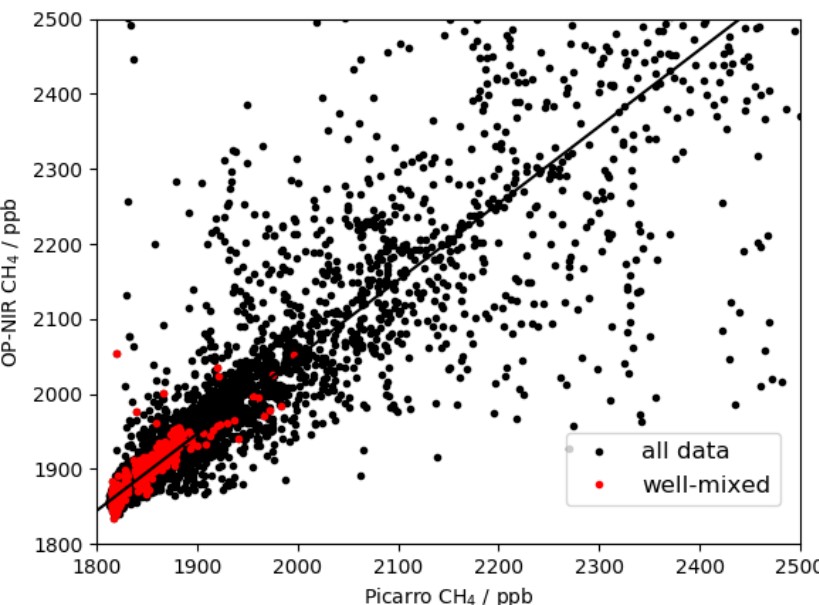

**Figure 8.** Comparison of path-averaged $CH_4$ measured by the open-path NIR and in situ $CH_4$ measured by the CRDS. All data points are shown in black, with those from well-mixed conditions (wind speeds greater than 2 m s$^{-1}$ and from the westerly sector) in red. The best fit line of OP-NIR = 1.0246 ± 0.0004 * Picarro for the fit of OP-NIR vs in situ measurements under well-mixed conditions is shown.





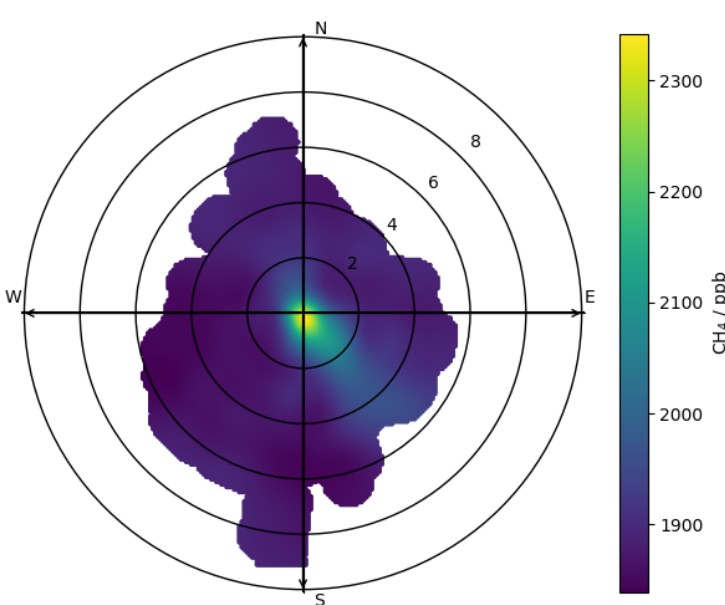

**Figure 9.** Polar bivariate plot showing the relationship between $CH_4$, wind speed and wind direction over the full campaign.



**Table 1.** Summary of the spectral windows used for retrieval of path-averaged concentrations in the open-path NIR spectra, see Griffith et al. (2018) for plots.

| Gas fitted | Interfering species | Spectral region / cm$^{-1}$ |
|:---:|:---:|:---:|
| $O_2$ | $H_2O$ | 7790 - 7960 |
| $CO_2$ | $H_2O$ | 4800 - 5050 |
| $CH_4$ | $H_2O$ | 5885 - 6150 |
| $H_2O$, HDO | $CO_2$ | 4910 - 5080 |
| CO | $H_2O$ | 4260 - 4310 |
| $N_2O$ | $CH_4$, $H_2O$ | 4300 - 4460 |



**Table 2.** Summary of open-path NIR instrument setup at EMAI. Previous work by Griffith et al. (2018) in Heidelberg is used as a reference.

| Measurement Period | Start Date | End Date | Pathlength (one-way) | Retroreflector | Telescope diameter |
|:---:|:---:|:---:|:---:|:---:|:---:|
| 1 | 20180804 | 20180809 | 600m | Hollow gold-coated | 10" |
| 2 | 20180809 | 20180816 | 600m | Solid uncoated BK7 glass | 10" |
| 3 | 20180816 | 20180831 | 1110m | Solid uncoated BK7 glass | 10" |
| 4 | 20180831 | 20181008 | 1500m | Solid uncoated BK7 glass | 10" |
| 5 | 20181018 | 20181212 | 1500m | Solid uncoated BK7 glass | 12" |
| ref | 201407 | 201411 | 1500m | solid UV quartz | 12" |



**Table 3.** Summary of the measurement performance during the field deployment at each pathlength and over changes in instrument setup. The numbered measurement periods and setups correspond to those in Table 2.

| Measurement period | Signal-to-noise ratio (SNR) | Repeatability ($1\sigma$) | | | |
|:---:|:---:|:---:|:---:|:---:|:---:|
| | | $CO_2$ / ppm | $CH_4$ / ppb | CO / ppb | $N_2O$ / ppb |
| 1 | 2050 | 0.66 | 8.2 | 6.9 | - |
| 2 | 6400 | 0.54 | 4.6 | 23.8 | 7.7 |
| 3 | 3750 | 0.35 | 3.7 | 26.9 | 35.5 |
| 4 | 2300 | 0.48 | 3.9 | 28.4 | 35.4 |
| 5 | 3200 | 0.26 | 2.0 | 17.0 | 21.7 |
| ref[#] | 750 | 1.7 | 21 | - | - |

[#] Deployment at Heidelberg (Griffith et al., 2018).



**Table 4.** Calculated biases of OP-NIR measurements relative to either in situ measurements or known mole fractions.

| Species | Ratio (mean ± s.d.) | r |
|---------|---------------------|------|
| $CO_2$  | $1.0267 \pm 0.0002$ | 0.91 |
| $CH_4$  | $1.0246 \pm 0.0004$ | 0.89 |
| $O_2$   | $1.0312 \pm 0.0001$ | N/A  |