# Peer review of "Performance of an open-path near infrared measurement system for measurements of CO2 and CH4 during extended field trials."

_Atmospheric Measurement Techniques, 2020_

## Referee Comment (RC1) · Anonymous Referee #1 · 13 Oct 2020

The authors present improvements and further characterization of an open-path NIR FTS system that was initially published (Griffith 2018). The improved system was operated over a 1.55 km path (one-way distance) with significantly improved precision compared to the initial system. The new system was characterized over several months in the field. Overall, the paper is well written and represents enough of an improvement in precision over the initial system that it is appropriate for AMT.

Specific Comments:

- Why was HITRAN 2008 used? How does the bias compare using 2016? What about using independently measured line strengths (e.g., Long et al, GRL, 2020,

doi:10.1029/2019GL086344)?

- What is the magnitude of the short-path correction? From Figure 2 it is apparent that the short-path + temperature effects combined are large, how much of this is attributable to the short-path correction? Related to this, how stable is the short-path spectrum? Did you, e.g., measure at the beginning and end of the campaign?

- Why was O2 not used to correct for pressure variations?

- The running mean of the well-mixed time series also removes potential sensor drift. Can you verify that the observed variability was indeed due to atmospheric variability and not drifting bias? Could you also add the Allan deviation plot?

- I didn't understand the argument for the SNR scaling with path length. If anything, I would think that turbulence should cause additional beam spreading and power loss. Could you explain that more? Do you know the size of the beam at the reflector array? If it is smaller than the array, could this explain it?

- There seem to be step changes in CO and N2O (near 2018-09 and 2018-10, respectively). Do you know what caused those?

- Did you correct the open-path measurements for water content?

- Do you see any difference in bias for the different path lengths?

- What is the expected sensitivity for CO and N2O with different reflectors?

Technical corrections:

- Lidar (DIAL and IPDA) should be added to the list of open path techniques. Probably this should be with TDLAS under a general heading of single-frequency laser techniques. In addition, the CLADS technique should be mentioned under this heading (e.g., Plant et al, Sensors, 2015 doi:10.3390/s150921315)

- The Queisser 2016 ref is DIAL, not DOAS.

[Figure]

- For the LIDAR refs, here is an IPDA system for CO2, CH4, and H2O: Wagner and Plusquellic, Appl Opt, 2016, doi:10.1364/AO.55.006292

- The Waxman 2017 ref should also be include in the frequency comb part of the introduction

- What was the size of the retro array?

- While similar to Griffith 2018, it would be nice to have the full spectrum shown here to provide clarity. It would also be nice to add the short-path spectrum to this figure.

- Could you label the direction towards potential sources on the polar CH4 plot?

- What averaging time was used for the precision numbers?

- Can you also add the O2 time series to Figure 5?

---

## Referee Comment (RC2) · Anonymous Referee #2 · 21 Dec 2020

This manuscript clearly presented the improvement of the instrumental system and setup on the measurement of CO2 and CH4, which are important to the global climate changes. In the introduction, authors did well in summarizing and comparing open-path measurement techniques, and at the end clearly pointed out that this work in on improvement of Griffith et al. (2018). In Section 2, it is also clear that what has been improved. This work also compared the open-path measurements with the in-situ CRDS measurements, quantified the difference, and provided potential reasons of the difference. The "Future Directions" nicely highlighted the potential their system of quantifying CO and N2O to apply to other environments. This content of this work in closely relevant to atmospheric trace gases observation techniques and GHG long term mon-

itoring, and suitable for publishing in AMT. The manuscript content is well organized. I would recommend publish after addressing some minor questions/revisions.

Line 99, about HITRAN database. Have you tried HITRAN2016 in your retrieval? Do you expect using HITRAN2016 will further improve your results?

Line 137 and Figure 2: the refitted black trace is also after excluding some poor measurements due to weather...as described in line 152-155, correct? So the difference between red and black is NOT purely due to the refit process, right? Since there are some low spikes in the "original", and they are gone in the "refitted". So please clarify this here in the text.

Line 140 and Figure 3: please include scale in the map, so readers know how far your site is from Sydney urban area.

Table 2 and Table 3 could merge into a single table. As the first time reader, I had a hard time remembering the difference between different periods, when reading Table 3.

Figure 4: It seems that there is some step change in N2O right before 2018-10. Do you know the reason? Did the retrieval of other gases changed at the same time of this step change in N2O?

Line 183, "a small improvement". Can you please put the percentage change after "a small improvement"?

Figure 7: although you have written the info in the text and figure caption, please label the Picarro, CSG, and the stock in Figure 7 to help readers and help your discussion on the following polar plot.

Line 220: " A polar bivariate plot of CH4..." you are talking about open-path measurement, correct? Here could be an confusion, since you just talked about Picarro measurement. So maybe include "from open-path measurement" after "CH4".

Figure 9: in the text, it is interesting to see the discussion of the enhancement of CH4 could relate to the cattle to the North, but the Figure 9 is very hard to tell if there is any enhancement to the North. So is it possible to further tune your color scale and somehow more clearly show the enhancement to the North.

Line 235: "There is much more scatter in the relationship," of what? I think it is also helpful to include r value of each fit in the text of this paragraph.

---

## Author Comment (AC1) · 23 Feb 2021

**Author response to Anonymous Referee #1.**

Nicholas M. Deutscher1, Travis A. Naylor1, Christopher G.R. Caldow1, Hamish L. McDougall1, Alex G. Carter1, and David W.T. Griffith1

1Centre for Atmospheric Chemistry, School of Earth, Atmospheric and Life Sciences, Faculty of Science, Medicine and Health, University of Wollongong, Wollongong, NSW, 2522, Australia

We thank Anonymous Referee #1 for their postive review of the manuscript. Below are our responses to their comments. Referee comments are *bold and italicised* with our responses below. Where it makes sense we have broken up the referee's specific comments into smaller chunks to hopefully make the responses clearer.

**5 1 Specific comments:**

**Why was HITRAN 2008 used?**

Here and in the past we have used HITRAN 2008 because of the speed of the computational calculations and working under the assumption that any biases, which might change with different spectroscopy, can be accounted for by corrections such as those detailed here.

10

**How does the bias compare using 2016?**

We've gone ahead and re-fitted some of the spectra using HITRAN 2016 instead of HITRAN 2008. While this process is still ongoing, initial results suggest that HITRAN 2016 makes a difference of approximately 1%, bringing the uncorrected (via comparison to in situ measurements) retrieved  $CO_2$  and  $CH_4$  closer to the in situ scale. We have re-fitted the period of overlap

15 between the open-path FTIR and in situ measurements, and are re-fitting the entire time series for all gases and will update the results with these.

In summary, HITRAN 2016 reduces the bias bewteen open-path FTIR and in situ measurements by about 50% for  $CO_2$  and  $CH_4$ , but has little effect for  $O_2$ . The 2.67% bias for  $CO_2$  is reduced to 1.41%, for  $CH_4$  the bias reduces to 1.61%, while for  $O_2$  it increases slightly to 3.24%. The uncertainties as assessed by the standard deviations barely change, but appear to be slightly bigher with HITRAN 2016 then HITRAN 2008 for  $CO_2$  and  $O_2$  and  $O_3$  and slightly reduced for  $CH_4$ .

20 higher with HITRAN 2016 than HITRAN 2008 for  $CO_2$  and  $O_2$  and slightly reduced for  $CH_4$

**What about using independently measured line strengths (e.g., Long et al, GRL, 2020, doi:10.1029/2019GL086344)?**

Thanks for bringing this paper (Long et al., 2020) to our attention, it's nice to know about. In this case, however, their work is not relevant as they re-measure linestrengths for the bands used by the Total Carbon Column Observing Network at 6228
cm-1 (30012 <- 00001) and 6348 cm-1 (30013 <- 00001), but here we use a stronger band near 4980 cm-1 (20012 <- 00001).</li>

**What is the magnitude of the short-path correction? From Figure 2 it is apparent that the short-path + temperature effects combined are large, how much of this is attributable to the short-path correction?**

Typically the short-path (hereafter 'SP') spectrum is about 1-2% of the intensity of the long-path (LP) spectrum. The SP 30 spectrum is essentially independent of the LP spectrum; that is, it simply adds to it.

A reference SP spectrum was recorded after any operator intervention or realignment. The relative intensity can vary, but the shape changes are minimal with time. A scaled reference SP spectrum is subtracted from each recorded spectrum, as described in section 2.1.1 (line 117). Its magnitude (or scaling factor) is determined for every measured spectrum. This is done by a least-squares fit of the reference SP spectrum in the 5000-5500 cm-1 region, where there is no contribution from the LP because it is

 $_{25}$  blacked out by  $H_2O$ . After subtraction, the corrected spectrum has zero intensity in the blacked out bands of the LP spectrum. This is illustrated in Figure 1.

The reduced continuum intensity after SP subtraction leads to an increased retrieved amount of trace gas, dependent on the relative contributions of the SP and LP spectra. This is because the trace gas absorptions retain the same depth in the corrected spectrum while the continuum decreases. For weaker bands, for example  $CH_4$ , the effect is approximately proportional to the SP/LP fraction, but amplied for stronger bands such as the  $CO_2$  bands at 4850 cm-1 and 4977 cm-1 in Figure 1.

The temperature correction, using the spectrum-fitted temperature rather than the measured temperature, usually decreases the retrieved mole fractions and therefore works in the opposite direction to the short-path correction. The fitted temperatures are typically 3-10 °C lower than locally measured, varying with time of day and solar radiation. The magnitude of this correction is influenced by two factors:

- 45 1. the lower measured air temperature leads to a higher calculated air density, and therefore a proportionally lower mole fraction of the trace gas for a given column of trace gas; and
  - 2. the retrieved column amount of the trace gas depends in a complex way on the temperature dependences of the absorption lines contributing to the spectrum.

In general, the spectroscopic effect is smaller than, but of the same sign as, the density effect.

- A series of figures show four retrievals for  $CO_2$  (Figure 2),  $CH_4$  (Figure 3) and  $O_2$  (Figure 4) for one day, October 22, 2018. The different retrievals are:
  - 1. the uncorrected retrieval using HITRAN 2008 (red);

40

- 2. the uncorrected retrieval using HITRAN 2016 (black);
- 3. the short-path corrected retrieval using HITRAN 2016 (green); and
- 4. the short-path and temperature-corrected retrieval using HITRAN 2016 (blue)

As mentioned in the response above, from HITRAN 2008 to HITRAN 2016, the 2.67% bias for  $CO_2$  is reduced to 1.41%, for  $CH_4$  the bias reduces to 1.61%, while for  $O_2$  it increases slightly to 3.24%, while the uncertainties do not appear to change.

**Why was $O_2$ not used to correct for pressure variations?**

60 If used,  $O_2$  would act as an alternative measure of the air density instead of using P/RT; however, the precision of the  $O_2$  measurement is less than that of the pressure and temperature. It therefore considerably increases noise in the derived trace gas mole fractions. Note this case is different from TCCON, where  $O_2$  is used to calculate the total air column.

**The running mean of the well-mixed time series also removes potential sensor drift. Can you verify that the observed variability was indeed due to atmospheric variability and not drifting bias? Could you also add the Allan deviation plot?**

On the time scales at which we have looked at data with the running mean, the bias relative to in situ measurements is constant within uncertainties.

Here we include some Allan deviation plots for the night of November 23-24, 2018 (17:15 - 04:30). For all gases, the determined Allan deviations are in agreement with the repeatability estimates given in Table 3 of the manuscript: 0.24 ppm (AD) vs 0.26 ppm for CO2 (Figure 5); 2.0 ppb vs 2.0 ppb for CH4 (Figure 6) 17.6 ppb vs 17 ppb for CO (Figure 7); and 22.2 ppb vs 21.7 ppb for N2O (Figure 8).

**I didn't understand the argument for the SNR scaling with path length. If anything, I would think that turbulence should cause additional beam spreading and power loss. Could you explain that more? Do you know the size of the beam at the reflector array? If it is smaller than the array, could this explain it?**

We propose the following re-wording of the paragraph at L187 in the article:

- For a (constant) detector-noise limited spectrum measurement, increasing pathlength from 600m to 1110m to 1500m
  would decrease spectrum signal and SNR as the inverse square of the pathlength, while in practice we observe an approximately inverse linear falloff. This is consistent with a component of noise proportional to signal, presumably due to a combination of turbulence and photon noise. The depth of absorption lines increases in proportion to pathlength, so the net absorption:noise ratio (and thus measurement precision) remains roughly constant with pathlength.
- 85 While we don't have an accurate measurement of the size of the beam at 1500 m, we observed that it was significantly larger than the retro-reflector array.

**There seem to be step changes in CO and N2O (near 2018-09 and 2018-10, respectively). Do you know what caused those?**

90 Yes, these were caused by author stupidity. The CO and  $N_2O$  weren't originally fitted for the period at 600 m one-way path, resulting in 2749 missing points at the start of their time series. However, in the plot displayed previously they were plotted as if they started at the same time as the other gases, and therefore the time axes for them are incorrect. This was addressed prior to all analysis, but somehow that figure was not updated. With some embarrassment, we've fixed this figure in the revised manuscript.

95

The step-change in  $N_2O$  therefore occurs over the data gap in October, when the system was modified to use a 12" telescope. The step-change in CO occurs when we shifted from 1110 to 1500 m pathlength. Both  $N_2O$  and CO are currently retrieved with limited signal and we are not sure what caused the step-changes, though in both cases the later periods appear to correspond to more realistic background southern hemisphere mole fractions of the gases.

100

**Did you correct the open-path measurements for water content?**

Yes, we did apply a correction to dry-air mole fractions to account for the retrieved  $H_2O$  content. We've clarified this in the revised manuscript at around line 108 by adding the text "The retrievals are corrected to dry-air mole fractions using the retrieved  $H_2O$ ."

**105**

**Do you see any difference in bias for the different path lengths?**

Unfortunately we do not have in situ data that overlap with the open-path measurement periods at different pathlengths, so we can't assess this. We have performed a preliminary assessment of the bias after setting the instrument up elsewhere over a 1.3 km one-way path and it is consistent with that determined here. Ideally we would perform co-located in situ measurements at

110 a variety of different pathlengths and locations (similar to the aircraft/AirCore comparisons done by TCCON) to determine the instrument's consistency between deployments and under a range of conditions.

**What is the expected sensitivity for CO and N2O with different reflectors?**

Excellent question! We haven't rigorously investigated this, but with some alternative reflectors that were lying around the lab
we see throughput at these frequencies of three times the current reflectors. This would presumably lead to at least improvement of √3 in the measurement repeatability for CO and N2O, to approximately 10 ppb and 13 ppb, respectively. Clearly this is not sufficient to be useful for detecting atmospheric signals except under exceptional circumstances; however, there are a couple of strategies that could be employed. We could either seek reflectors designed for maximum signal at this range and create a retro-reflector array alternating reflectors with peak performance for CO/N2O and CO2/CH4, and/or further increase the size
of the current array to allow for more returning signal at pathlengths where the beam overfills the current array size. For all situations, we need to refine our setup perform further tests to assess these possibilities.

**2 Technical corrections:**

Lidar (DIAL and IPDA) should be added to the list of open path techniques. Probably this should be with TDLAS under a
general heading of single-frequency laser techniques. In addition, the CLADS technique should be mentioned under this heading (e.g., Plant et al, Sensors, 2015 doi:10.3390/s150921315)

We've refined the section discussing open-path technologies following these suggestions and included reference to Plant et al. (2015) and Nikodem et al. (2015).

**130 The Oueisser 2016 ref is DIAL, not DOAS.**

Noted and corrected. Thanks.

**For the LIDAR refs, here is an IPDA system for CO2, CH4, and H2O: Wagner and Plusquellic, Appl Opt, 2016, doi:10.1364/AO.55.006292**

135 Thanks, we have included this (Wagner and Plusquellic (2016)).

**The Waxman 2017 ref should also be include in the frequency comb part of the introduction**

We have made reference to this paper (Waxman et al. (2017)) in the introduction as well.

**140 What was the size of the retro array?**

The size of the final array was 600 (h) x 625 (w) mm. We've added this information to the manuscript around line 88: with a total dimension of 600 mm by 625 (height x width).

**While similar to Griffith 2018, it would be nice to have the full spectrum shown here to provide clarity. It would also be 145 nice to add the short-path spectrum to this figure.**

The full spectrum is shown below in Figure 9, along with the short-path spectrum and a zoom (x20) of the short-path spectrum for clarity. We've added this figure to the revised manuscript with reference to it at the end of the section describing the shortpath correction (around line 122, section 2.1.1).

**150 Could you label the direction towards potential sources on the polar CH4 plot?**

We've avoided this to try to limit misinterpretation of the polar plot radial axis as being distance rather than wind speed. Many of the potential sources are also about the same distance from the open-path as the pathlength itself, and therefore marking them on this polar plot is potentially confusing. Instead, we have marked the location of CSG well and grazing areas on the map in Figure 7. The updated Figure 7 (now Figure 8) from the manuscript is shown below in Figure 10. We also added text to the manuscript referring to the indicated sources at the end of the paragraph discussing the polar bivariate plot (around line 226).

155

**What averaging time was used for the precision numbers?**

The averaging time varied depending on the period used. In each case we picked a period where the atmosphere was wellmixed and there were no obvious changes in trace gas mole fractions due to local sources. These correspond to the periods in 160 Table 1.

**Can you also add the O2 time series to Figure 5?**

We don't have an in situ measurement of  $O_2$ , so we've not added this to that particular time series plot. The  $O_2$  mole fraction should not vary significantly from 0.2096.

165

**References**

- Long, D., Reed, Z., Fleisher, A., Mendonca, J., Roche, S., and Hodges, J.: High-Accuracy Near-Infrared Carbon Dioxide Intensity Measurements to Support Remote Sensing, Geophysical Research Letters, 47, e2019GL086344, https://doi.org/10.1029/2019GL086344, https://onlinelibrary.wiley.com/doi/10.1029/2019GL086344, 2020.
- 170 Nikodem, M., Plant, G., Sonnenfroh, D., and Wysocki, G.: Open-path sensor for atmospheric methane based on chirped laser dispersion spectroscopy, Applied Physics B: Lasers and Optics, 119, 3–9, https://doi.org/10.1007/s00340-014-5938-3, https://link.springer.com/article/ 10.1007/s00340-014-5938-3, 2015.
  - Plant, G., Nikodem, M., Mulhall, P., Varner, R., Sonnenfroh, D., and Wysocki, G.: Field Test of a Remote Multi-Path CLaDS Methane Sensor, Sensors, 15, 21 315–21 326, https://doi.org/10.3390/s150921315, http://www.mdpi.com/1424-8220/15/9/21315, 2015.
- 175 Wagner, G. A. and Plusquellic, D. F.: Ground-based, integrated path differential absorption LIDAR measurement of CO\_2, CH\_4, and H\_2O near 16 μm, Applied Optics, 55, 6292, https://doi.org/10.1364/ao.55.006292, https://www.osapublishing.org/ viewmedia.cfm?uri=ao-55-23-6292{&}seq=0{&}html=truehttps://www.osapublishing.org/abstract.cfm?uri=ao-55-23-6292https: //www.osapublishing.org/ao/abstract.cfm?uri=ao-55-23-6292, 2016.

Waxman, E. M., Cossel, K. C., Truong, G.-W., Giorgetta, F. R., Swann, W. C., Coburn, S., Wright, R. J., Rieker, G. B., Coddington, I.,

180 and Newbury, N. R.: Intercomparison of open-path trace gas measurements with two dual-frequency-comb spectrometers, Atmospheric Measurement Techniques, 10, 3295–3311, https://doi.org/10.5194/amt-10-3295-2017, https://www.atmos-meas-tech.net/10/3295/2017/, 2017.

**Figure 1.** Long-path spectrum in the 4700 - 5600  $\text{cm}^{-1}$  region before (red, as measured) and after (blue, corrected) correction for the contribution from a short measurement path resulting from reflection directly off the secondary mirror.

---

## Author Comment (AC2) · 23 Feb 2021

**Author response to Anonymous Referee #2.**

Nicholas M. Deutscher[1], Travis A. Naylor[1], Christopher G.R. Caldow[1], Hamish L. McDougall[1], Alex G. Carter[1], and David W.T. Griffith[1]

[1]Centre for Atmospheric Chemistry, School of Earth, Atmospheric and Life Sciences, Faculty of Science, Medicine and Health, University of Wollongong, Wollongong, NSW, 2522, Australia

We thank Anonymous Referee #2 for their postive review of the manuscript. Below are our responses to their comments. Referee comments are ***bold and italicised*** with our responses below.

***Line 99, about HITRAN database. Have you tried HITRAN2016 in your retrieval? Do you expect using HITRAN2016 will further improve your results?*** As detailed in the response to Anonymous Referee #1, we had not previously tried HITRAN2016 for our retrievals but have now done so. Using HITRAN2016 instead of HITRAN2008 improves the derived biases compared to the in situ measurements by about 50% for $CO_2$ and $CH_4$ and makes little difference to $O_2$ or the derived repeatabilities for any of the gases.

***Line 137 and Figure 2: the refitted black trace is also after excluding some poor measurements due to weather...as described in line 152-155, correct? So the difference between red and black is NOT purely due to the refit process, right? Since there are some low spikes in the "original", and they are gone in the "refitted". So please clarify this here in the text.*** Actually the data shown in Figure 2 is filtered identically for both the original and refitted data shown, that is the differences are indeed entirely due to the refitting process. The largest differences that are referred to in the referee's comment usually occur at times when the long-path signal is low, and therefore the short-path reflections make a large contribution. As discussed in the response to Anonymous Referee #1, the effect of the short-path correction is to increase the retrieved trace gas mole fractions - the larger the relative contribution of the short-path, the larger the trace gas increase. This is what is seen here.

***Line 140 and Figure 3: please include scale in the map, so readers know how far your site is from Sydney urban area.*** The site is 52 km from Sydney CBD. We've added this to the text and included a scale bar on the map in Figure 3. This required changing from a Google maps image to one generated using Google Earth

***Table 2 and Table 3 could merge into a single table. As the first time reader, I had a hard time remembering the difference between different periods, when reading Table 3.*** We considered merging these into a single table when originally drafting the publication, but opted to keep them separate to avoid including too much information in a single table. For that reason we've kept them separate, but added a column to Table 3 to summarise the instrument setup. Hopefully this makes it clearer without cluttering up a single table too much. A copy of

the revised Table 3 is shown below (Table 1)

30     *Figure 4: It seems that there is some step change in N2O right before 2018-10. Do you know the reason? Did the retrieval of other gases changed at the same time of this step change in N2O?*

As noted in the response to Anonymous Referee #1, this step-change should actually occur over the gap during which the instrument didn't operate while the telescope was being changed from 10" to 12" diameter version. The step-change therefore corresponds to the change due to this instrument modification.

35

    *Line 183, "a small improvement". Can you please put the percentage change after "a small improvement"?*

Sure. The changes are 18% for $CO_2$ and 44% for $CH_4$. For CO the repeatability is about 3.5 times worse because of the changed reflector substrate and poorer reflectivity in the CO retrieval region. We've changed the text in the manuscript to: **and a small improvement in trace gas retrieval repeatability for $CO_2$ (18%) and $CH_4$ (44%)**

40     *Figure 7: although you have written the info in the text and figure caption, please label the Picarro, CSG, and the stock in Figure 7 to help readers and help your discussion on the following polar plot.*

We have marked the location of the Picarro, coal-seam gas well, and areas in which we observed grazing stock.

    *Line 220: " A polar bivariate plot of CH4..." you are talking about open-path measurement, correct? Here could be*
45 *an confusion, since you just talked about Picarro measurement. So maybe include "from open-path measurement" after "CH4".*

Thanks. Yes, this is from the open-path measurements, and we have clarified this as suggested.

    *Figure 9: in the text, it is interesting to see the discussion of the enhancement of CH4 could relate to the cattle to the*
50 *North, but the Figure 9 is very hard to tell if there is any enhancement to the North. So is it possible to further tune your color scale and somehow more clearly show the enhancement to the North.*

Actually this is quite difficult to do; however, we've included here a version of the plot (Figure 1) that only shows the northerly data at wind speeds above 0.75 m/s, which hopefully illustrates a little better that there are some small enhancements from the north and slightly west of north. We've not included this in the manuscript.

55

    *Line 235: "There is much more scatter in the relationship," of what? I think it is also helpful to include r value of each fit in the text of this paragraph.*

Here we are referring to the in situ versus open-path comparison for $CH_4$ compared to $CO_2$. We've clarified this by adding the text **for $CH_4$ than $CO_2$**. We've added the r values to the text as well.

60

**References**

Griffith, D. W. T., Pöhler, D., Schmitt, S., Hammer, S., Vardag, S. N., and Platt, U.: Long open-path measurements of greenhouse gases in air using near-infrared Fourier transform spectroscopy, Atmospheric Measurement Techniques, 11, 1549–1563, https://doi.org/10.5194/amt-11-1549-2018, https://www.atmos-meas-tech.net/11/1549/2018/, 2018.

[Figure]

**Figure 1.** Polar bivariate plot showing the relationship between CH$_4$, wind speed and wind direction over the full campaign for winds with a northerly component and wind speeds greater than 0.75 m/s.

**Table 1.** Summary of the measurement performance during the field deployment at each pathlength and over changes in instrument setup. The numbered measurement periods and setups correspond to those in Table 2 from the manuscript.

| Measurement period | Instrument setup (path, reflector, telescope) | Signal-to-noise ratio (SNR) | Repeatability ($1\sigma$) | | | |
|---|---|---|---|---|---|---|
| | | | $CO_2$ / ppm | $CH_4$ / ppb | CO / ppb | $N_2O$ / ppb |
| 1 | 600m, gold, 10" | 2050 | 0.66 | 8.2 | 6.9 | - |
| 2 | 600m, glass, 10" | 6400 | 0.54 | 4.6 | 23.8 | 7.7 |
| 3 | 1110m, glass, 10" | 3750 | 0.35 | 3.7 | 26.9 | 35.5 |
| 4 | 1500m, glass, 10" | 2300 | 0.48 | 3.9 | 28.4 | 35.4 |
| 5 | 1500m, glass, 12" | 3200 | 0.26 | 2.0 | 17.0 | 21.7 |
| ref[#] | 1500m, quartz, 12" | 750 | 1.7 | 21 | - | - |

[#]Deployment at Heidelberg (Griffith et al., 2018).